# Thermal Comfort Analysis Using System Dynamics Modeling—A Sustainable Scenario Proposition for Low-Income Housing in Brazil

Cylon Liaw [1,*], Vitória Elisa da Silva [2], Rebecca Maduro [1], Milena Megrè [1], Julio Cesar de Souza Inácio Gonçalves [2], Edmilson Moutinho dos Santos [1] and Dominique Mouette [3]

1 Institute of Energy and Environment, University of São Paulo, São Paulo 05508-010, Brazil
2 Institute of Technological and Exact Sciences, Federal University of Triângulo Mineiro, Uberaba 38064-200, Brazil
3 School of Arts, Sciences and Humanities, University of São Paulo, São Paulo 03828-000, Brazil
* Correspondence: cylon.liaw@usp.br

**Abstract:** As a riveting example of social housing in Brazil, the Minha Casa Minha Vida program was set in 2009 to diminish the 6-million-home housing deficit by offering affordable dwellings for low-income families. However, recurrent thermal discomfort complaints occur among dwellers, especially in the Baltimore Residential sample in Uberlândia City. To avoid negative effects of energy poverty, such as family budget constraints from the purchase of electric appliances and extra costs from power consumption, a simulation based on system dynamics modeling shows a natural ventilation strategy with a mixed combination of sustainable and energy-efficient materials (tilting window with up to 100% opening, green tempered glass, and expanded polystyrene wall) to observe the internal room temperature variation over time. With a 50% window opening ratio combined with a 3 mm regular glass window and a 12.5 cm rectangular 8-hole brick wall, this scenario presents the highest internal room temperature value held during the entire period. From the worst to the best-case scenario, a substantial reduction in the peak temperature was observed from window size variation, demonstrating that natural ventilation and constructive elements of low complexity and wide availability in the market contribute to the thermal comfort of residential rooms.

**Keywords:** system dynamics; thermal comfort; Minha Casa Minha Vida; natural ventilation; bioclimatic architecture; social housing; energy poverty

## 1. Introduction

In Brazil, energy consumption in the residential sector relies on electricity, wood, and liquefied petroleum gas (LPG), with a crescent use of natural gas, according to the Brazilian Energy Balance—2020 report [1]. By 2020, this particular sector consumed 10.8% of the total energy supply in Brazil, only ahead of the agribusiness and services sectors. Figure 1 illustrates how these energy resources have slightly changed over the last decade, with a strong presence of renewable energy share (67%) due to electricity generation based on renewable resources such as hydro, solar, wind, and biomass.

Moreover, due to the LPG price soaring between 2016 and 2019, wood became a direct substitute for fossil fuel for cooking, especially among low-income families. Further on, with a global economy aggravated by the COVID-19 pandemic's resonating effects, it is expected that wood consumption remains this increasing trend until Brazilian families' purchasing power is regained. In this sense, Brazil's Energy Research Office (EPE) had earlier pointed out this direction and showed a 1.8% growth in wood use in 2020 when compared to the previous year [1]. Additionally, a greenhouse gas (GHG) emission rise related to this expansion is likely to occur. In Brazil, anthropogenic $CO_2$ emissions associated with the residential sector accrued 19.4 Mt $CO_2$-eq in 2020 [1]. Though it represented close to 5%

of Brazil's total emissions in the corresponding year, we highlight that the country held 12th place in the global emission ranking [2]; in other words, a relatively small percentage denotes a robust addition to the global warming threat; therefore, it is a relevant volume not to be ignored.

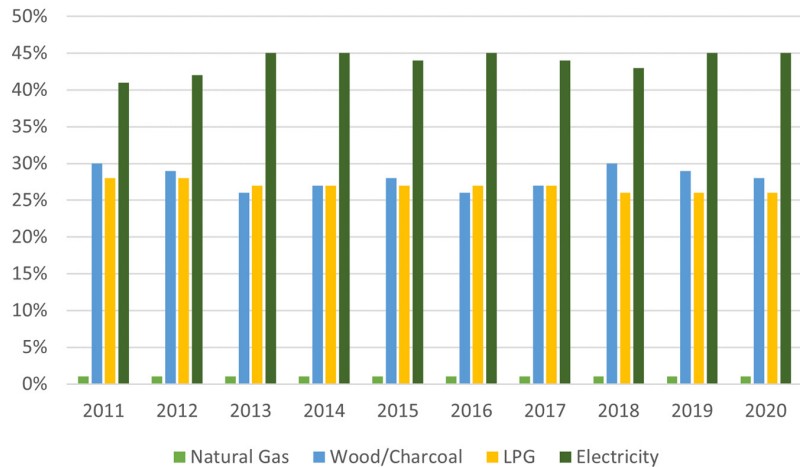

**Figure 1.** Residential sector energy consumption by source (source: authors' own elaboration from [1] database).

Although subject to successive price rises during the same period, electricity consumption likewise kept continuously increasing. A perceived 4.05% growth between 2019 and 2020 resulted from a combination of diversified factors in the presence of the COVID-19 pandemic, such as higher absence from work, extended home office conditions, federal financial aid, and home appliance acquisition [1]. The latter creates a demand for better convenience associated with energy efficiency measures, as indicated by Lamberts et al. [3], especially for thermal comfort as a determinant of human well-being and when it comes to prolonged indoor stays [3,4].

Throughout history, humanity has sought comfort, protection, and safety, with modern societies ending up primarily in urban concentrations and spending most of their day indoors. Nevertheless, according to Felix et al. [5], when faced with a poorly designed environment, the consequences for humans living under such conditions could represent, for example, loss of productivity and health problems due to inadequate indoor air quality and lack of thermal comfort.

Regarding thermal comfort, it encompasses a whole range of data types and collection methods [4], including physiological and psychological aspects, ambient factors [6], whether from active and controlled technologies (through ventilation, heating, and air conditioning), architecture (materials, building design, and landscaping), and even activities and clothing, as shown in Daghigh's [7] and Mallick's [8] articles. In essence, each residence presents a particular mix toward adaptation to the regional climate and according to available alternatives in the market.

The role of climate in defining the conditions and thermal performance of indoor environments has become more relevant. In a study carried out in Portugal in 2017 [9], a high mortality rate related to exposure to excessive cold was identified. This result signaled the impact of energy poverty during a certain season of the year, the urgency of proposing solutions for this impacted layer of society, and the pressure put on the income of these families. Simões and Leder [10] describe energy poverty as a hurdle to accessing energy services, often threatening families, for instance, in tropical climates with exposure to internal overheating due to the lack of cooling solutions.

Additionally, Almeida et al. [11] present important findings about the particular discussion related to families' income versus the necessity to improve thermal comfort in new homes, while observing the impact on carbon emissions. This time, research

conducted with a family building in Porto, Portugal, explored the relationship between the embodied carbon emissions and embodied energy of the materials used, showing a mutual increase associated with each renovation package to this building [11]. The results show that while the use of renewable energy can be positive in terms of reducing the use of non-renewable primary energy, the embodied energy and implicit life-cycle emissions associated with the materials and processes applied to building solutions can potentially harm their sustainability [11]. This relationship between social housing and thermal comfort issues was further developed in [12–17].

In this sense, natural ventilation is an effective option that relies on natural airflow to assure appropriate thermal conditioning of the environment, which provides favorable comfort conditions to its occupants and improvement of indoor air quality with no income pressure [18]. However, over the last decades, buildings have adopted new technologies to provide such coziness, often depending on electric appliances, and gradually both natural ventilation and lighting strategies were put aside [18]. This could represent an increase in energy consumption for families who are often unable to pay for the energy to maintain a comfortable standard of living at home [19].

When it comes to home appliances used in Brazil for this matter, a survey regarding possession and habits of use of electrical equipment in the residential class [20] indicated that almost 76% of the sample maintained a ventilator or air circulator in residences, whereas approximately 17% owned an air conditioner [20]. Additionally, according to this survey conducted between 2018 and 2019, air conditioners were mostly held by the upper classes with the highest income strata, while other appliances were equally distributed among all classes [20]. It has been observed that the energy used for thermal comfort in buildings follows a rising trend not only in Brazil but also worldwide: between 1990 and 2016, the power demand for space cooling more than tripled [21].

With these findings, this paper sheds light on thermal comfort as a lingering issue in social housing in Brazil, considering that families with financial constraints can barely afford more efficient ventilating appliances, i.e., air conditioners, and are incapable of withstanding a higher electricity bill within a narrow budget (energy poverty) [10].

As a riveting example of social housing, the Minha Casa Minha Vida (MCMV) program set in 2009 represented the Brazilian national push to diminish a 6-million-home housing deficit, equivalent to 10.2% of the total private-owned houses, by offering affordable dwellings for low-income families. However, Fundação João Pinheiro's report [22] showed some flaws concerning the program's family selection process focused on income and putting other relevant preconditions aside, such as existing precarious housing, cohabitation of families, and excessive financial burden with renting [22]. For this reason, the MCMV program arguably contributed to reducing the housing deficit a decade later, as this figure remained close to its initial level and has been stable since 2016, though this deficit decreased in relative numbers (dropped from 10.2% to 8%) as indicated by the official MCMV assessment report [23].

To enable such an ambitious housing program, a maxed-out number of residences would have to be built with a limited capital volume. Not surprisingly, a low investment allocated to each house resulted in poor construction quality and energy efficiency measures [24]. As a consequence of inadequate building design that disregards bioclimate zone peculiarities, the lack of thermal comfort remains a resonating complaint among MCMV residents. According to Baltimore Residential's occupants, a condominium located in Uberlândia City, Minas Gerais state (Figure 2), "tight, hot, and stuffy" senses describe the indoor apartment characteristics, as indicated in the post-occupation assessment report [25], due to the inefficient airflow often trapped in the center hall and only single-sided ventilation, as seen in Figure 3 [25].

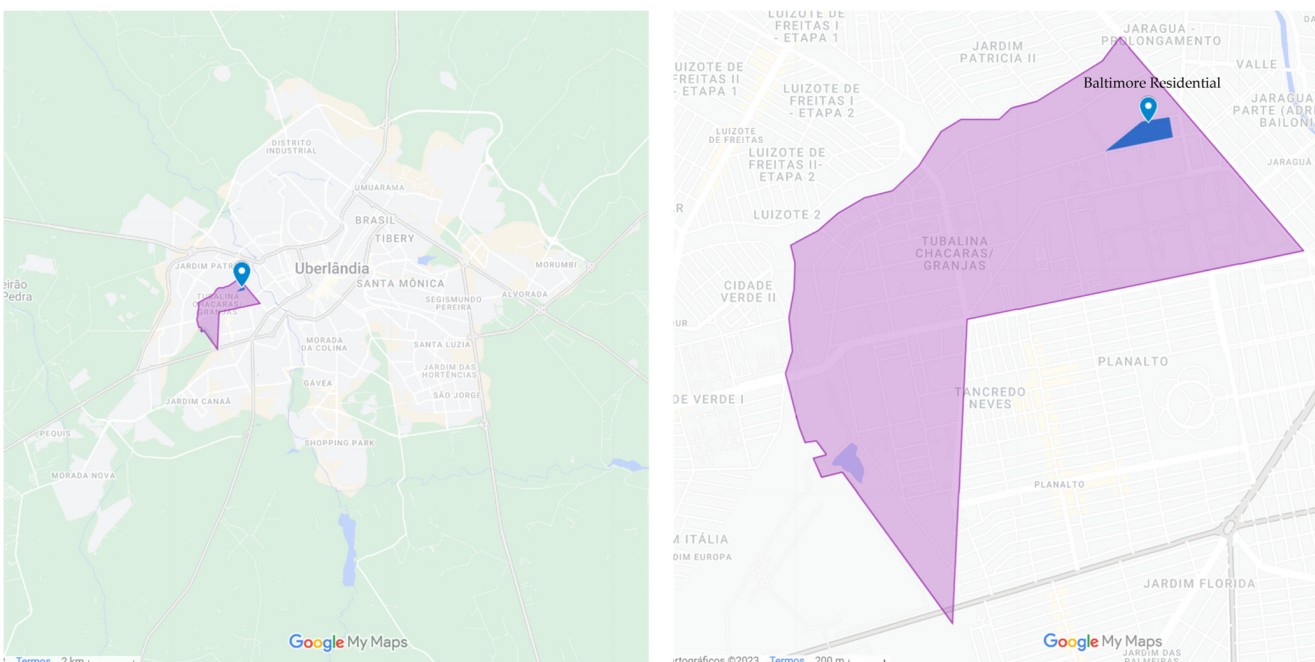

**Figure 2.** Baltimore Residential's location in Uberlândia, Brazil (source: authors' own elaboration on Google My Maps).

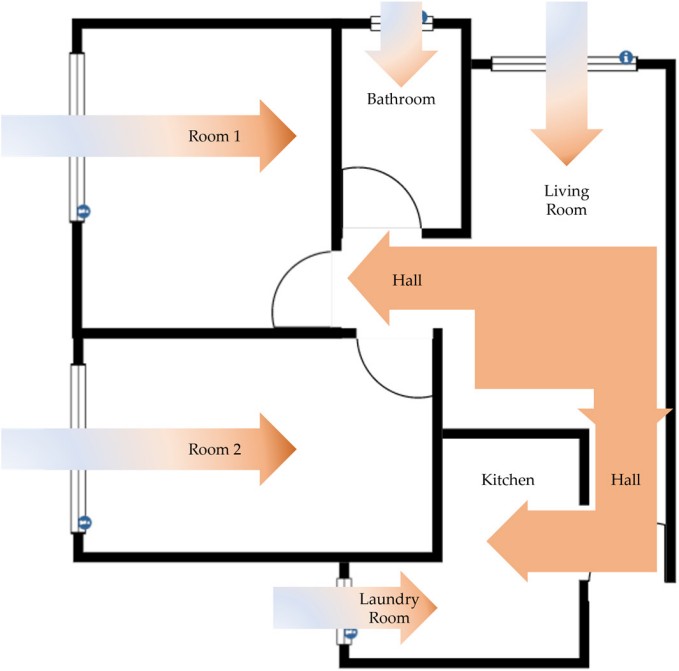

**Figure 3.** Airflow in a Baltimore Residential apartment (source: authors' own elaboration based on [25] data and created on SmartDraw website).

Most MCMV homes are "one-size-fits-all" units that share a standardized size, room allocation, and building design, and this rigid structure limits any adaptation to local climate conditions and hence replicates thermal comfort issues previously described [25,26]. For this reason, social housing in Brazil presents no significant architectural pattern differences, leading to extra expenses for further renovation in an attempt to mitigate thermal discomfort [25].

For the above-mentioned reasons, this paper will take a single room from Baltimore Residential to run a system dynamics (SD) model based on the literature review's sugges-

tions on sustainable solutions for thermal comfort improvement [27,28]. The SD model should shed light on the underlying cause-and-effect relationships that could explain why a particular behavior occurred in the first place, simulating the causality between factors through positive and negative feedback [27]. By applying this tool, it is possible to deal with high-order non-linear problems and other complex systems; therefore, it has been used by researchers to investigate, for example, green buildings [29] and construction management [28] promotion strategies.

The simulation goal is to verify possible benefits to indoor temperature using a mix of passive design strategies on windows, walls, and glasses. As an example, modifying window type and openings directly contribute to suitable ventilation, as opposed to the standard sliding type window seen in Brazilian social housing projects, which reduces the effective area of ventilation by 50% [30].

With these low-cost modifications applied during the construction phase, it is highly expected that dwellers will rely on alternative forms of ventilation instead of electric appliances, thus contributing to lesser carbon emissions without compromising the family budget. As described by Geels et al. [31], the transition toward a sustainable energy system includes:

> "( . . . ) major changes in buildings, energy and transportation systems to substantially increase energy efficiency, reduce demand or imply a shift from fossil fuels to renewable supplies. These transitions imply not only technical changes, but also changes in consumer behavior, markets, institutions, infrastructure, business, models, and cultural discourses."

Throughout the literature review, thermal comfort in social housing has promoted various discussions regarding its potential solutions to particular climate characteristics. Based on bioclimatic architecture premises, this paper considers the local climate zone to determine the type of adapted architecture with no energy consumption and a low ecological footprint [32]. In this sense, natural ventilation is deemed to provide adequate thermal comfort and indoor air quality, relying on the building design and its interaction with the local environment [33].

Bodach and Hamhaber [24] performed a compelling simulation regarding cost assessment with an energy efficiency focus on a social housing project located in Rio de Janeiro. The Mangueira project shares Baltimore's building design and, therefore, those same flaws are present in both contexts. The authors presented an economic evaluation highlighting an additional cost according to the proposed change and potential savings from energy-efficient appliances, though they have not simulated to what extent temperature would fall due to these modifications.

Simões et al. [34] pointed out significant shortcomings due to unguided house renovation and expansion, mostly resulting in thermal discomfort and consequent unhealthy living conditions in low-income houses. According to the authors, residents in such an environment would be forced to use adaptative strategies to minimize discomfort, leading to high-demanding energy alternatives. To avoid such risks and unnecessary spending for these particular units, this paper will not consider applying the simulated mix of modifications in a post-occupation phase, but rather in the initial building plan. Just like Baltimore's complaints, Simões et al. [34] gathered similar opinions in João Pessoa, also identifying other Brazilian cases related to thermal issues in low-income housing in Campina Grande and Pato Branco, but did not carry out any simulation regarding passive ventilation mechanisms and related temperature drops.

One of the latest literature reviews focused on the MCMV program considered an entire decade from 2009 to 2019, with Bavaresco et al. [35] collecting relevant information regarding energy performance in Brazilian social housing. Though not proposing any solution for thermal comfort dissatisfaction nor running any simulation, the authors exposed the most impacting factors for energy efficiency in this particular matter within a total of 93 national publications, which matched the corresponding predefined research parameters.

The authors have researched studies focused on themes related to thermal comfort, natural ventilation, bioclimatic architecture, social housing, and energy poverty. A significant foundation on thermal comfort and natural ventilation in tropical regions such as Brazil could be identified through this research [10,36], which was reinforced by articles that consider, in addition to thermal comfort, issues related to social housing [10,37]. The researched works on the application of natural ventilation techniques in different bioclimatic zones indicated guidelines regarding the use of scenarios involving the configuration of buildings, wind direction, and efficiency as important factors for the application of natural ventilation solutions [38]. Yet, the authors of [39,40], dealing with social housing and energy poverty, focused on cooling solutions in social housing to minimize the thermal discomfort caused by high temperatures and reduce energy poverty in the considered regions. In addition, such articles corroborate with other studies carried out regarding the aforementioned themes, opening the way for new works with solutions that are more tailored to the experienced bioclimatic zones, allowing their improvement and application for families exposed to such climatic conditions [41].

As mentioned before, natural ventilation and sustainable insulated walls are shown as low-cost alternatives to diminish thermal discomfort. In addition, the authors highlighted that a single solution would not be possible as every locality shows a very unique set of bioclimatic aspects; moreover, given their specificities, even if two or more municipalities belong to the same bioclimatic zone, they may differ in construction aspects and should receive specific guidance [35]. Dorsey et al. [42] led a study on the use of natural building methods and how these could be incorporated into buildings and improve community development, land use planning, and architectural design as well as issues related to climate change. Another example draws attention to the use of straw bales in buildings and their proper adaptation to the official construction code, which could be promoted on a large scale [43]. These relevant discussions may also be replicated in the Brazilian MCMV context to promote low-carbon construction materials.

In conclusion, with the great variety of regional scenarios and the respective bioclimatic zones, much of the existing literature for low-income thermal comfort focused on describing related issues and listing possible alternatives, rather than simulating potential temperature drops linked to these solutions. This paper proposes a novel structured method based on SD to test some of the cited alternatives for Baltimore Residential's single rooms to increase thermal comfort.

## 2. Materials and Methods

The following tasks will be carried out to support social housing programs by providing thermal comfort to residents, with adequate thermal performance and energy efficiency.

(a) Detailing the parameters that interfere with thermal comfort;
(b) Comparing the thermal performance of current scenarios and regulatory parameters;
(c) Proposing minimum constructive requirements to be adopted without changing any architectural design.

This article presents the development of a model capable of predicting the internal thermal sensitivity of an individual room in Baltimore Residential using the SD method, which is a computer-aided approach for stock and flow diagram strategy development and better decision-making in dynamic and complex systems [44]. In this case, the software Vensim (from Ventana Systems, Inc.; Salisbury; UK) was used for the simulation exercise based on feedback theory, which complements system thinking approaches.

For the model creation, three steps were essential to evaluating and prioritizing the most suitable projects that are relevant to communities on a social, environmental, and economic level, according to Castrillon-Gomez et al. [45]:

1. Problem identification: how to improve thermal comfort, knowing in advance the constructive elements that influence the thermal energy flow of a certain volume of social coexistence;

2.  Hypothesis planning: by researching solutions that needed minimum construction requirements, given the availability of these materials and the complexity of installation;
3.  SD model construction: through identification of state variables (internal energy of the room and internal temperature of the room) and considered relationships between parameters (percentage of window opening, solar factor of the glass, thermal transmittance of the wall).

Further on, the conclusion of the stock and flow diagram considers the addition of auxiliary variables, which will be further explained in this paper. These values remain the same in all scenarios when only the solar factor of the glass, wall thermal transmittance, and room window opening percentage parameters are changed to simulate the behavior of the system over time. In the computational model, the constructive parameters of an individual room in Baltimore Residential are also transcribed. After this data collection, the elements of the system are interrelated through mathematical equations. Ultimately, changing values of the constructive elements of the system results in a scenario variety with different indoor thermal sensitivity of the room.

*2.1. Selection Criteria for Building Materials*

A myriad of building materials is currently available to provide a desirable level of thermal comfort. Nevertheless, it comes with a broad array of costs, with top-tier technologies requiring higher expenditure [46]. Considering the MCMV program, the use of poor-quality materials has contributed to general dissatisfaction [25,35]. Although dealing with a restricted budget, the federal housing program may rely on a better material selection that does not necessarily reflect higher costs. In this sense, the following items were chosen from the literature reviewed in the introduction section [24,33–35], especially concerning cost/benefit [11], thermal efficiency [37,38,40], and national availability [10] aspects:

(a)  A tilting window that allows up to 100% opening of its usable area—Baltimore Residential's rooms are single-sided ventilation examples; therefore, there is only one facade to explore the natural ventilation possibility. Letting airflow without losing privacy is an advantage, which depends on the difference between the indoor and outdoor temperatures and the window opening percentage [47]. In this paper, a tilting window will provide a half or total opening ratio in substitution to the traditional sliding type often found in MCMV housing. In addition to this, according to [48,49], the window size and location should contribute to thermal comfort; however, each municipality has its own construction code and there is no general guidance regarding window location in Uberlândia City [50]. For this reason, only the window size will be considered for the simulation in accordance with local regulations.
(b)  A wall with EPS (expandable polystyrene) monolithic panel—deemed as a lightweight and low-cost alternative for thermal and acoustic insulation, with the benefit of being water-resistant, the EPS panel has shown great performance while covered by cement plaster layers. Although it is non-biodegradable, this fossil fuel product can be recycled to contribute to mitigating its production carbon footprint and enabling more sustainable construction [51]. Compared to other traditional wall materials, such as hollow ceramic bricks or concrete blocks, expandable polystyrene presents one of the lowest thermal conductivities, which means less heat is transmitted from one side of the wall to another [52].
(c)  A window with green tempered glass—instead of 3 mm regular glass, which greatly contributes to thermal dissatisfaction due to a high thermal transmittance—a fair cost/benefit solution is found with green tempered glass. With the addition of ferrous oxide and ferric oxide in different concentrations, a range of glass colors is available (including greenish ones) to absorb and reduce incoming solar infrared heat [53]. The 3 mm tinted glass provides 1/3 less solar radiant heat energy that enters the room (solar factor) when compared to regular glass [54]. In this case, according to the building material prices list used in the MCMV program [55], the selected 6 mm

green tempered glass presents much higher protection for a reasonable cost, and is considered an optimized solution among distinct possibilities, especially regarding budget restraints in the national program. The simulation with green tempered glass is only carried out when the window is half open, as in this situation the glass is an element that participates in the heat flow. However, when the window is completely open, the glass is out of the picture and does not affect the system.

## 2.2. Calculation of Natural Ventilation

According to Lamberts, Dutra, and Pereira [3], natural ventilation through unilateral openings is an energy-efficient method to adjust the thermal comfort of buildings. Thus, it is necessary to understand the climatic conditions of the region, such as wind speed and average temperature. The wind speed, provided by meteorological stations, must be corrected for the height of interest, also as a function of the distance between houses [3]. In addition, the wind pressure coefficients of the region must be observed. From this information, the average corrected wind speed must then be calculated [3], as given by Equation (1):

$$\text{Vcorrected} = \text{Vaverage} \times \text{F}_t \times \left(\text{H}^{\text{Fr}}\right) \tag{1}$$

Vcorrected: average corrected wind speed (m/s);
Vaverage: average annual wind speed measured by the weather station (m/s);
H: building ridge height (m);
$\text{F}_t$: topographic factor;
Fr: terrain roughness factor.

The average annual wind speed gauged by the weather station in the Uberlândia region is close to 1.67 m/s [56]. According to Lamberts, Dutra, and Pereira [3], the topographic and terrain roughness factors are, respectively, 0.35 and 0.25. Moreover, as reported by Villa, Saramago, and Garcia, the building ridge height is equivalent to 15 m [25].

Window opening is pivotal to estimating the corresponding airflow in the room [3]. Depending on the window opening ratio, its useful area may vary and, in this case, a 50% or 100% opening is considered. Regarding Baltimore Residential's rooms, only single-sided ventilation is available, which directly contributes to the air flux amount carried into the room.

$$Q = 0.025 \times a \times \text{Vcorrected} \tag{2}$$

Q: airflow with natural ventilation in the room ($m^3$/s);
a: opening window useful area ($m^2$);
Vcorrected: average corrected wind speed (m/s).

The living room window has a total area of 1.78 $m^2$ [25]. To ensure the air quality of a room, a minimum number of air changes per hour is defined. As airflow is given in cubic meters per second, the result is multiplied by 3600 to establish the number of air changes per hour, according to Equation (3) [3]:

$$n = \left(\frac{Q \times 3600}{v}\right) \tag{3}$$

n: number of air changes per hour;
Q: airflow with natural ventilation in the room ($m^3$/s);
v: ventilated room volume ($m^3$).

According to the floor plan of the Baltimore Residential apartment evaluation (Figure 3), the room has an area equivalent to 10.95 $m^2$ and a ceiling height equal to 2 m, thus the volume of the room is equal to 21.9 $m^3$ [25]. The average annual temperature of the Uberlândia region is approximately 22.1 °C [57]. Moreover, the internal room temperature becomes the desired variable, in which values vary when modifying the constructive parameters of the environment.

After determining the number of air exchanges for the room, which depends on the tightness of the air openings, it is important to understand that this exchange will translate into a heat removal in the room. To calculate the thermal load, the sensible heat must be determined, which is related to the temperature difference between the interior and exterior, as shown in Equation (4) [3]:

$$Q_{SE} = m_{air} \times c_{air} \times \left( \frac{n \times v}{3600} \right) \times \Delta T \tag{4}$$

$Q_{SE}$: sensible heat (W);
$m_{air}$: air density (1.2 kg/m$^3$);
$c_{air}$: specific heat of the air (1000 J/kg·°C);
n: number of air changes per hour;
v: ventilated room volume (m$^3$);
$\Delta T$: internal and external temperature difference (°C).

Heat flux represents the heat transfer rate through a unit area section of the window and it is uniform (invariant) across the entire area of the window opening as well as the glass. The heat loss through the window area is given by Equation (5) [58]:

$$\phi = \frac{Q_{SE}}{a} \tag{5}$$

$\phi$: heat flux removed from the room (W/m$^2$);
$Q_{SE}$: sensible heat (W);
a: opening window useful area (m$^2$).

### 2.3. Calculation of the Internal Thermal Energy of the Room

To understand which material has better thermal performance, it is necessary to know its thermal transmittance index, which means the amount of heat in watts that passes from one surface of the wall to another, per square meter at the degree of variation of temperature between the surfaces [47].

In this project, two types of envelopes are simulated: a 12.5 cm 8-hole brick wall and an EPS (expandable polystyrene) monolithic panel. The thermal transmittance of the first material is 2.94 W/m$^2$·°C [3], while the second is 0.42 W/m$^2$·°C [59]. From these indexes, it is possible to calculate the heat flux of Baltimore Residential's rooms, as given by Equation (6) [3]:

$$q = U \times \Delta T \tag{6}$$

q: added heat flux in the room (W/m$^2$);
U: thermal transmittance (W/m$^2$ °C);
$\Delta T$: internal and external temperature difference (°C).

It is understood that Equation (6) calculates the heat flux added in the room, due to the thermal transmittance of the wall, while Equation (5) calculates the heat flux removed from the room, determined by the number of air changes through the window. The difference between both added and removed heat flux from the room results in the internal thermal energy of the room. This calculation is better understood by noting that the room is considered to be a stock where energy accumulates. From the initial internal energy value at a given time "t", it is known that what is added to the stock between time t and the next point in time, denoted by "t + 1", is also added to obtain the stock value at the, "t + 1" point, given by Equation (7) [60,61]:

$$E(t) = E(t_0) + \int_{t_0}^{t} [(q \times A) - (\phi \times a)] \tag{7}$$

E: internal energy of the room (J);

q: added heat flux in the room ($W/m^2$);
A: area of the wall where solar radiation falls on ($m^2$);
$\phi$: heat flux removed from the room ($W/m^2$);
a: ventilation area of the living room window ($m^2$).

Thus, it is concluded that the room ambiance works as heat storage, where the subtraction of the inlet and outlet flows results in the internal temperature of the room, given by Equation (8) [60]:

$$Tinternal = \frac{E}{c_{air} \times m_{air} \times v} \tag{8}$$

Tinternal: internal room temperature (°C);
$c_{air}$: specific heat of air at constant pressure (1000 J/kg·°C);
$m_{air}$: air density (1.2 $kg/m^3$);
v: ventilated room volume ($m^3$).

*2.4. Model Construction*

A causal diagram depicts a reduced representation of causal links and mathematical equations to facilitate the understanding of the variables' evolution and their interconnected rationale. As can be seen in Figure 4 (made with the Vensim software PLE 9.0.1x64 version), there are two negative balancing loops (B1/B2), whose functions are to smooth the internal thermal energy inside the room, which is directly related to the internal room temperature. Loop B1 is related to the heat flux that passes through the wall and window glass, while Loop B2 considers the heat flux removed from the room by the number of air changes through the window opening.

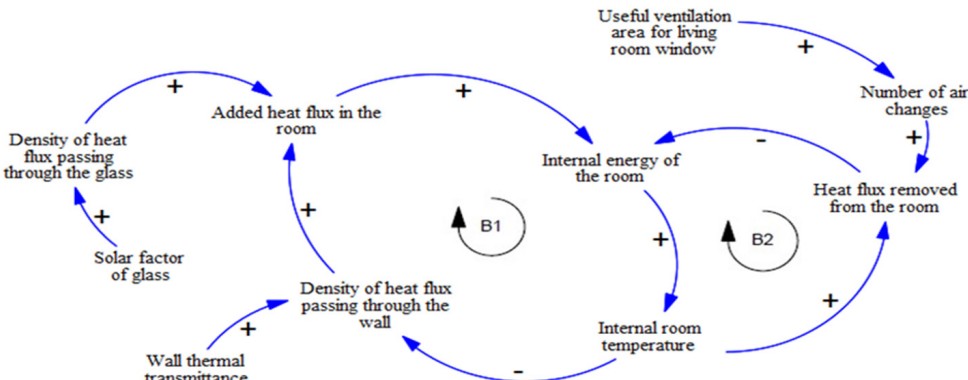

**Figure 4.** Causal diagram for Baltimore Residential's rooms (source: authors' own elaboration using the Vensim software).

To develop a stock and flow diagram (SFD), four building blocks are used, according to Ahmad et al. [62]: stock, flow, and auxiliary variables, and also a connector. A stock shows an accumulation of any variable; in the case under study, energy is the variable accumulated in the room. The flow is attached to a stock and responsible for increasing or depleting the stock level, with the variable "added heat flux in the room" as the input and the variable "heat flux removed from the room" as the output. Auxiliary variables can be parameters or values calculated from other system variables. All auxiliary variables and their values can be seen in Table 1. Finally, a connector or an arrow denotes the connection and control between the system variables. Ultimately, the four building blocks are displayed within the stock and flow diagram, as seen in Figure 5.

**Table 1.** List of auxiliary variables and their respective values (source: authors' own elaboration).

| Auxiliary Variables | Value | Reference |
|---|---|---|
| Average wind speed | 1.67 m/s | [63] |
| Terrain roughness factor | 0.25 | [3] |
| Topographical factor | 0.35 | [3] |
| Ventilation area of the living room window | 1.78 m$^2$ | [25] |
| Building ridge height | 15 m | [25] |
| Wind pressure coefficient for unilateral ventilation | 0.025 | [3] |
| Initial temperature | 22.1 °C | [63] |
| Specific heat of air | 1000 J/kg·°C | [60] |
| Area of the wall where solar radiation falls on | 5.27 m$^2$ | [25] |
| Absorptivity of the external surface of the wall | 0.2 | [3] |
| External surface thermal resistance of the wall | 0.04 | [3] |
| Room volume | 21.9 m$^3$ | [25] |

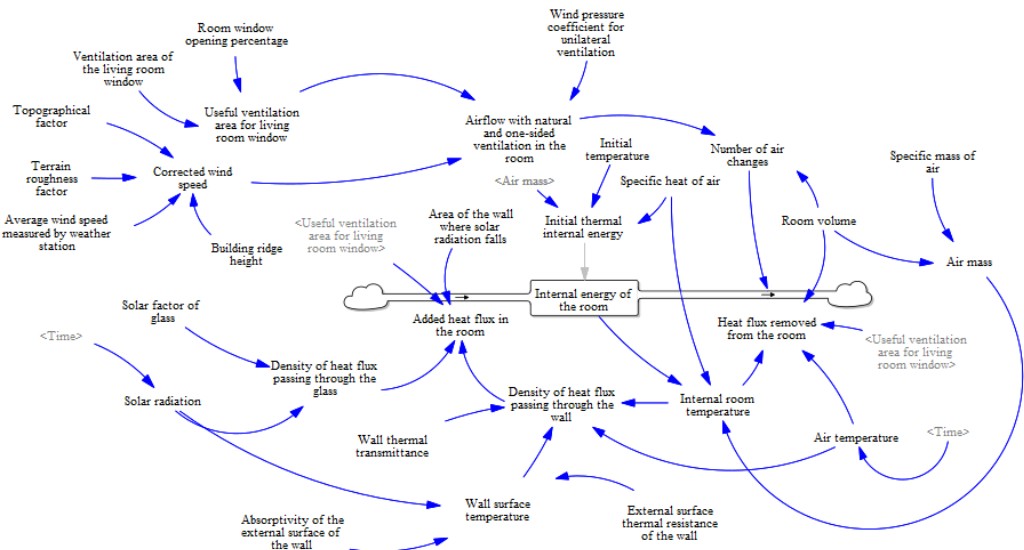

**Figure 5.** Stock and flow diagram for Baltimore Residential's rooms (source: authors' own elaboration using the Vensim software).

## 3. Results and Discussion

The thermal comfort evaluation in indoor environments is commonly performed according to the American Society of Heating, Refrigerating and Air Conditioning Engineers (ASHRAE) standard ASHRAE 55 [64], which has been largely explored in the literature [65–71] and defines indoor operating temperatures ranging from 23.0 °C to 26.0 °C (for 35% relative humidity) and from 22.5 °C to 25.5 °C (for 65% relative humidity). In the Uberlândia region where this simulation was carried out, the relative humidity is above 65% [63]. For analysis purposes, a temperature of 25.5 °C was adopted as the thermally comfortable value, being the maximum value according to its operating range.

For the model purposes, a 5-month period between October 2021 and February 2022 taken from the National Institute of Meteorology (INMET) database [63] contained significant data on the temperature of Uberlândia during the hottest months of 2021/2022 time frame. Bringing a harsh-condition sample to an extensive but not exhaustive simulation would reflect how far thermal discomfort has long been affecting Baltimore Residential's occupants, which is critical to inflict health disorders, especially in the considered period. Moreover, the potential benefits of the building materials combination would also be stressed to the fullest.

From 3000 temperature samples registered at least twice per day (also at night), it was noticed that 33% remained above ASHRAE's thermally comfortable threshold value

of 25.5 °C. In this sense, it was reasonable to extract a random day from this selection for the simulation.

The model also considers the average solar radiation in the region and the outdoor air temperature, which were obtained from the National Institute of Meteorology (INMET), specifically from the Uberlândia station [63]; in Figure 6, the solar radiation presents a low level at the beginning of the day, eventually increasing at times close to noon, and decreasing at the end of the afternoon. On the other hand, on this particular day, between 10h00 and 15h00, the outdoor temperature kept rising even with less solar radiation, also seen in Figure 6. These data are displayed over time, between 06h00 and 18h00, considering the solar radiation occurrence.

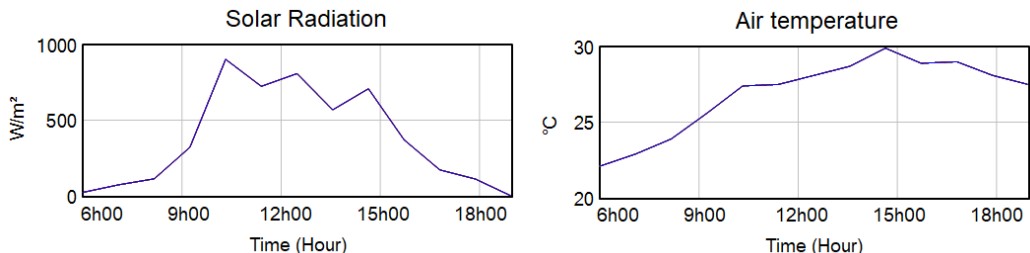

**Figure 6.** Regional solar radiation and outdoor air temperature (source: authors' own elaboration using the Vensim software).

After incorporating the data into the model, the next step requires establishing values for the input variables such as useful ventilation area, glass material, and wall material, which make up each scenario, as shown in Table 2.

**Table 2.** Technical specifications of simulated scenarios (source: authors' own elaboration).

| Scenario | Window Opening (%) | Window Glass Material | Wall Material |
|---|---|---|---|
| 50% Window Opening | 50 | 3 mm regular glass | 12.5 cm rectangular 8-hole brick |
| 50% Window Opening + EPS | 50 | 3 mm regular glass | EPS monolithic panel |
| 50% Window Opening + Green Tempered Glass | 50 | 6 mm green tempered glass | 12.5 cm rectangular 8-hole brick |
| 50% Window Opening + Green Tempered Glass + EPS | 50 | 6 mm green tempered glass | EPS monolithic panel |
| 100% Window Opening | 100 | - * | 12.5 cm rectangular 8-hole brick |
| 100% Window Opening + EPS | 100 | - * | EPS monolithic panel |

* Once the window is completely open, the window glass material does not affect the system.

As an outcome obtained from the simulation, once initially set with the average annual temperature of the region (22.1 °C) at 06h00, the internal room temperature variation over time is represented in Figure 7, in which is possible to identify three local maxima similar to Figure 6, since it is directly linked to solar radiation.

With a 50% window opening ratio combined with a 3 mm regular glass window and a 12.5 cm rectangular 8-hole brick wall, this scenario presents the highest internal room temperature value held during the entire period, which illustrates the thermal discomfort peak pointed out by Villa, Saramago, and Garcia [25]. Even with the substitution of window glass, wall materials, or both for allegedly more sustainable and efficient ones, while remaining with a 50% window opening ratio, there was only a small drop in perception regarding the internal room temperature.

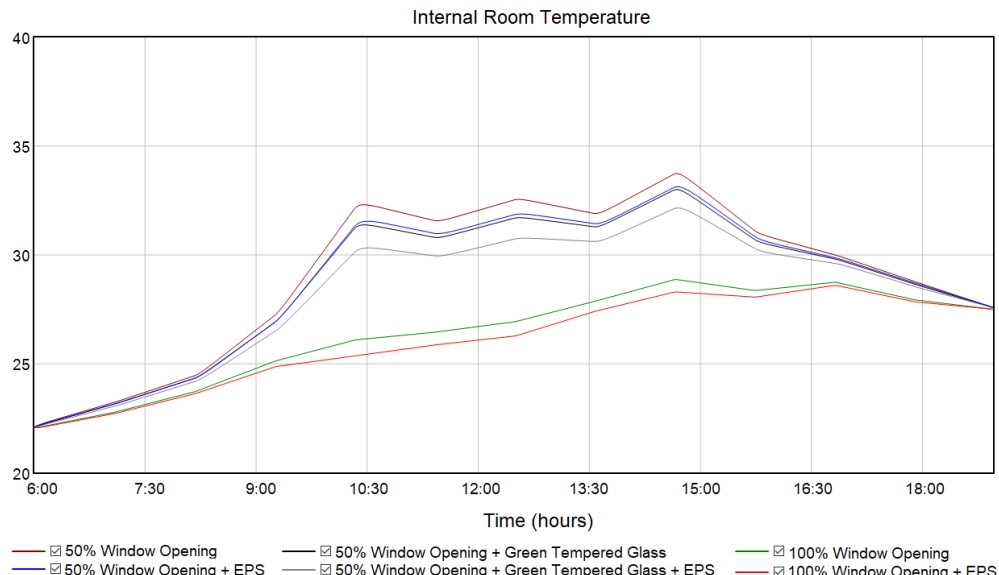

**Figure 7.** Internal room temperature variation over time—scenario performance (source: authors' own elaboration using the Vensim software).

On the other hand, the lowest temperatures over time were achieved by applying a fully open window (100% opening ratio), regardless of its material, with an additional reduction when used alongside an EPS wall. In this context of a wide-open window, it is worth mentioning that the window glass material did not impact the simulation since heat flow and solar radiation had no obstacles whatsoever. It is noticed that, even with the maximum outdoor temperature shown at 15h00, the room remained at a lower temperature.

The generated results in Figure 8 show the amount of time (as a percentage of the full period) during which each scenario exhibited temperature values above 25.5 °C.

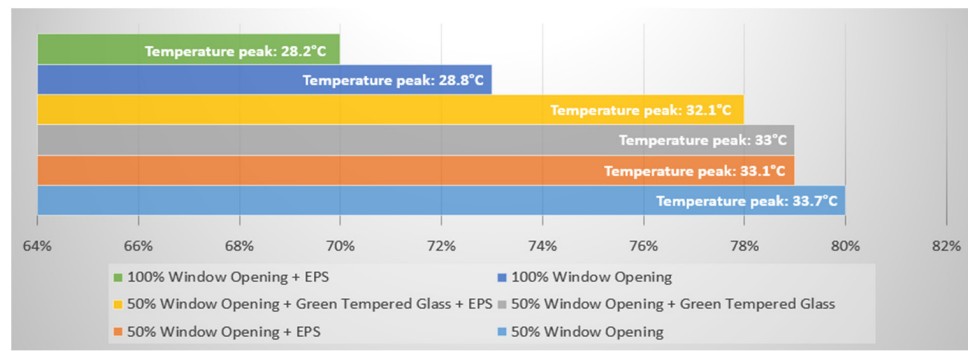

**Figure 8.** Percentage of time in which the internal room temperature was above 25.5 °C (source: authors' own elaboration).

Regarding the worst-case scenario, it is verified in the corresponding situation that the internal room temperature remains above 25.5 °C for the longest time, equivalent to 80% of the considered interval, peaking at 33.7 °C close to 15h00. The other three scenarios that also include a 50% window opening have lower percentages, with a temperature drop at the decimal order, including:

(+) EPS wall: 0.6 °C temperature peak reduction and 79% of the time with an internal room temperature above 25.5 °C;

(+) Green tempered glass: 0.7 °C temperature peak reduction and 79% of the time with an internal room temperature above 25.5 °C;

(+) Green tempered glass and EPS wall: 1.6 °C temperature peak reduction and 78% of the time with an internal room temperature above 25.5 °C

Indeed, the 100% window opening ratio contributes to the greatest sensitivity to the system, exerting a strong influence on the heat flux removal as heat change frequency increases. Other elements such as the solar factor of the glass and thermal transmittance of the wall influence the amount of energy added to the environment. As heat struggles to enter the room, it also struggles to leave it, meaning that the impact of both does not significantly alter the internal temperature of the room.

The best scenario encompasses a fully opened window and an EPS wall, with the temperature staying above 25.5 °C for 70% of the time, peaking at 28.2 °C, which is still above the operative range for thermal comfort. However, it is considered that by applying this scenario to the entire residence, it would be possible to offer a thermally comfortable environment.

In addition to these results, the predicted mean vote index (PMV) method from standard 55-2004 of the ASHRAE [64] can be applied to determine the thermal comfort of the room. The PMV index is used to measure the level of thermal comfort, as shown in Table 3.

**Table 3.** PMV index scales with thermal perception (source: authors' own elaboration).

| Scales | Thermal Perception |
|--------|--------------------|
| 3 | Hot |
| 2 | Warm |
| 1 | Slightly warm |
| 0 | Neutral |
| −1 | Slightly cool |
| −2 | Cool |
| −3 | Cold |

The results of the scenarios taken from the SD model were analyzed according to the ASHRAE 55 thermal comfort PMV method using the CBE Thermal Comfort Tool [72,73], developed at the University of California at Berkeley. Table 4 shows the summary of the input values of the factors used in the method.

**Table 4.** Parameters used to calculate the thermal comfort indices (source: authors' own elaboration).

| PMV Index Factors | Inputs | Reference |
|-------------------|--------|-----------|
| Airspeed | 1.67 m/s | [63] |
| Relative humidity | 72% | [63] |
| Metabolic rate | Standing, relaxed: 1.2 MET | [25] |
| Clothing level | Knee-length skirt, short-sleeve shirt, sandals, underwear: 0.5 clo for summer | [74] |

The inputs in Table 4 remain the same for all scenarios, only the internal temperature of the room varies according to each scenario, and the temperature value used was the peak temperature in order to analyze the most extreme situation presented by the SD model. For the PMV analysis purpose using the CBE Thermal Comfort Tool, the operative temperature used is equal to the peak temperature without variation over time.

As the window opening factor leads to a compelling contribution to the temperature drop, modifying the window size may also bring an additional benefit when combined with the previous construction elements [48,49], following the same rationale presented in Equation (2), in which the airflow volume is directly related to the useful area of the window. According to Uberlândia's Municipal Construction Code [50], each window area should be at least equivalent to 50% of the required illuminated area, which is 1/6 of the total room area. In this case, a minimum window area of 0.9125 m$^2$ complies with the local regulation; it is worth mentioning that the previous results were obtained from a window area equal to 1.78 m$^2$.

A new simulation for the internal room temperature regarding different sizes for the window was performed, ranging from 1.00 to 3.00 m$^2$ (the minimum window area of 0.9125 m$^2$ was rounded up for better comparison). Figure 9 shows the internal room peak temperature encompassing the previous six scenarios for different window areas.

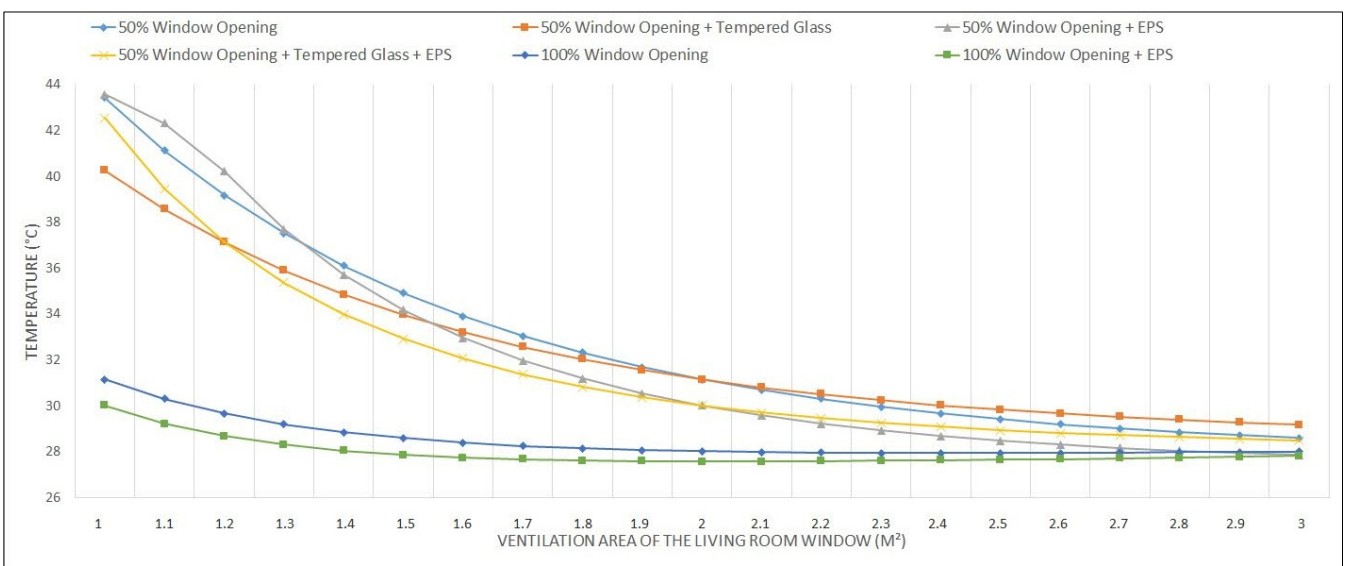

**Figure 9.** Internal room peak temperature behavior according to the window area variation (source: authors' own elaboration).

Compared to the MCMV standard window size (1.78 m$^2$), it is observed that the smallest window area (1.00 m$^2$) provides the highest peak temperature in the room for all scenarios, as a direct result of less airflow volume and air exchange. In this case, a smaller window can raise the internal room peak temperature close to 11 °C ("50% Window Opening", "50% Window Opening + EPS", and "50% Window Opening + Tempered Glass + EPS" scenarios). With a 2.00 m$^2$ window area, a temperature decrease of up to 3 °C is detected in the "50% Window Opening + EPS" scenario, while others remain lower than a 1 °C drop. When a 3.00 m$^2$ window area is applied, almost 5 °C is withdrawn from the room temperature, shown in the "50% Window Opening" and "50% Window Opening + EPS" scenarios.

Both the "100% Window Opening" and "100% Window Opening + EPS" scenarios demonstrated a slight temperature drop for 2.00 m$^2$ and 3.00 m$^2$ window areas in comparison to the standard MCMV window measure (1.78 m$^2$), whereas the smallest window area provided an addition of up to 3 °C to the peak temperature. Moreover, it was noticed that as the window area gets closer to 3.00 m$^2$, the internal room peak temperature difference between the simulated scenarios reaches a maximum value of 1.37 °C. In conclusion, there is a temperature threshold that limits the benefits from the window area increase, which decision-making should take into account when determining building material costs and building structure safety. Table 5 contains the PMV results from the CBE Thermal Comfort Tool [73].

Since the smallest window area (1.00 m$^2$) brings the highest internal room temperature for all the simulated scenarios, it is also expected to show a PMV index that translates to thermal discomfort. In this case, a PMV index above 3 (hot) is shown for all four scenarios that include "50% Window Opening", whereas the "100% Window Opening" scenarios stay within the neutral scale (close to 0).

**Table 5.** PMV results for each scenario and window area (source: authors' own elaboration).

| Scenario | Ventilation Area of the Room Window 1.00 m² | | Ventilation Area of the Room Window 1.78 m² | | Ventilation Area of the Room Window 2.00 m² | | Ventilation Area of the Room Window 3.00 m² | |
|---|---|---|---|---|---|---|---|---|
| | Peak Temperature (°C) | PMV Index | Peak Temperature (°C) | PMV Index | Peak Temperature (°C) | PMV Index | Peak Temperature (°C) | PMV Index |
| 50% Window Opening | 43.43 | 6.22 | 33.70 | 1.73 | 31.16 | 0.81 | 28.60 | −0.03 |
| 50% Window Opening + EPS | 43.57 | 6.29 | 33.10 | 1.50 | 30.02 | 0.43 | 27.87 | −0.26 |
| 50% Window Opening + Green Tempered Glass | 40.26 | 4.62 | 33.00 | 1.47 | 31.16 | 0.81 | 29.19 | 0.15 * |
| 50% Window Opening + Green Tempered Glass + EPS | 42.54 | 5.77 | 32.10 | 1.14 | 30.02 | 0.43 | 28.51 | −0.06 * |
| 100% Window Opening | 31.16 | 0.81 | 28.80 | 0.03 * | 28.02 | −0.21 * | 28.01 | −0.22 |
| 100% Window Opening + EPS | 30.02 | 0.43 | 28.20 | −0.16 * | 27.58 | −0.35 | 27.82 | −0.28 |

* Lowest PMV values among all scenarios.

For the MCMV standard window area (1.78 m²), the analysis results show a PMV index close to Scale 2 (warm), while the most favorable scenario (100% Window Opening + EPS) places between Scale 0 (neutral) and Scale -1 (slightly cool). As can be seen, a fully opened window has a slight difference either combined with an EPS wall or a brick wall alternative, nevertheless, both scenarios have a larger distinction when it comes to compared to half-opened-window scenarios, which is an indication that wide-open windows are relevant to reaching the thermal comfort goal. With a larger window area (2.00 m² and 3.00 m²), a greater airflow volume and more frequent air exchanges are key to improving thermal comfort: the previous shows PMV values between 1 (slightly hot) and 0 (neutral), whereas the latter resides mostly between 0 (neutral) and -1 (slightly cold). Interesting to notice that the "50% Window Opening" scenario benefits the most from the adoption of a larger window, which demonstrates the potential gains from the window area variation on the internal room temperature.

## 4. Conclusions

This research demonstrates that natural ventilation and the use of constructive elements of low complexity and wide availability in the market (tilting window with up to 100% opening, green tempered glass, and EPS wall) can contribute to the thermal comfort of a residential room. From the worst to the best-case scenario, a substantial reduction in the peak temperature was obtained from window size variation, without any use of electrical equipment, such as a ventilator or air conditioning. In addition, this paper shed light on solutions such as natural ventilation and more sustainable and energy-efficient building materials that would not incur power consumption, considering that the majority of electric appliances are not affordable for most of the low-income population, which would also increase the household electricity costs, but mainly, elevate anthropogenic greenhouse gas emissions.

Moreover, the fact that obtaining a significant decrease in temperature just by fully opening the window or modifying its size makes the solution more tangible, given the ease of its implementation, it is something that can be done even in existing homes through a well-planned renovation. Serving as a baseline to encourage continuous field development in new buildings and renovations [11], it will also avoid any increase in electricity consumption during the operational phase for buildings, considering their entire life cycle [10].

The EPS wall has other advantages in addition to thermal comfort, as it is a light material and easy to install, and also makes construction more agile; therefore, it is recommended and its effect on thermal comfort remains positive. The green tempered glass alternative should be analyzed in terms of costs and benefits compared to the tilting window with 100% opening; if its cost is considerably higher, it is worth installing the full opening window or a larger one.

The effect caused by green tempered glass and EPS wall in hampering the heat entrance is important, but it should be noted that the heat flow exit also depends on the same window opening, thus the prioritized constructive element should be the 100% opening tilting window, which also assures the privacy of residents.

In conclusion, the advancement of constructive technologies and materials used in civil construction significantly contributes to expanding access to more efficient buildings in the thermal field without increasing electrical consumption. The system dynamics method may be further adapted and replicated to other circumstances, considering various locations and different realities among low-income housing in Brazil. Ultimately, managing thermal comfort for a national-size social housing program means adding significant value to the quality of life for millions at a reasonable cost. For policymakers, social housing should not remain only a matter of using low-cost materials but elevating their benefits to the fullest on a sustainable path.

**Author Contributions:** Conceptualization, C.L., V.E.d.S. and R.M.; methodology, V.E.d.S. and J.C.d.S.I.G.; software, V.E.d.S. and J.C.d.S.I.G.; validation, D.M. and J.C.d.S.I.G.; formal analysis, C.L., V.E.d.S. and R.M.; investigation, C.L., V.E.d.S., R.M. and M.M.; resources, C.L., V.E.d.S., R.M. and M.M.; data curation, C.L. and V.E.d.S.; writing—original draft preparation, C.L., V.E.d.S. and R.M.; writing—review and editing, C.L., V.E.d.S. and J.C.d.S.I.G.; visualization, C.L. and V.E.d.S.; supervision, D.M. and J.C.d.S.I.G.; project administration, D.M.; funding acquisition, E.M.d.S. and J.C.d.S.I.G. All authors have read and agreed to the published version of the manuscript.

**Funding:** We gratefully acknowledge the support of the RCGI—Research Centre for Greenhouse Gas Innovation, hosted by the University of São Paulo (USP) and sponsored by FAPESP—São Paulo Research Foundation (2014/50279-4 and 2020/15230-5) and Shell Brasil, and the strategic importance of the support given by ANP (Brazil's National Oil, Natural Gas and Biofuels Agency) through the R&D levy regulation. This study was financed in part by the Coordenação de Aperfeiçoamento de Pessoal de Nível Superior—Brasil (CAPES)—Finance Code 001. This work is based upon financial support from Conselho Nacional de Desenvolvimento Científico e Tecnológico (CNPq - 407631/2021-6).

**Institutional Review Board Statement:** Not applicable.

**Informed Consent Statement:** Not applicable.

**Data Availability Statement:** Not applicable.

**Acknowledgments:** Special recognition to Samantha Maduro, whose valuable architectural experience within the Minha Casa Minha Vida program shed light on sustainable materials and state-of-the-art practices.

**Conflicts of Interest:** The authors declare no conflict of interest. The funders had no role in the design of the study; in the collection, analyses, or interpretation of data; in the writing of the manuscript; or in the decision to publish the results.

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
