# Peer review of "Thermal Comfort Analysis Using System Dynamics Modeling—A Sustainable Scenario Proposition for Low-Income Housing in Brazil"

_sustainability, doi:10.3390/su15075831_

Round 1

Reviewer 1 Report

The current Manuscript (hereinafter referred to as MS) is about to analyze the effects of natural ventilation and two basic components of a building structure on the indoor temperature. The analysis was carried out on the apartment of a condominium located in Uberlândia city, in Brazil. The main goal of the MS is clear, and the research topic is relevant, however, major revision of the MS is proposed because of the followings.

Major comments:

1. The key publications regarding the research topic shall be referred in the MS. (In the current MS around 45% of the 39 references are technical reports or other online sources.) There is no discussion in the MS since the results are not interpreted from the perspective of previous studies. Identifiers/links of the references [1], [3], [5], [10], [12], [13], and [15] are missing, while I can’t identify reference [20]. Please refer the key publications in the introduction and add a proper discussion to the MS.

2. In the following cases please add proper references to the MS which support the Authors’ statements:

-  in line 69 (the Authors mentioned clothing in connection with thermal comfort without references)

-  in lines 75-77

-  in line 81 (a survey is mentioned without the reference)

-  in lines 224-225

-  Please add the source of the solar radiation data to the MS.

-  In lines 393-394 some papers shall be referred, in which the thermal comfort evaluation in indoor environments is performed according to the American standard ASHRAE 55.

3.  It is not obvious what literature is mentioned by the Authors:

-  in lines 116-118,

-  in lines 221-222,

-  in lines 224-225,

-  in line 293.

4.  In the following cases further explanations are needed:

-  In lines 199-200 what kind of project is mentioned by the Authors?

- In line 212 "secondary variables and parameters" are mentioned by the Authors. Please list them.

5. Please add reference and proper description to the System Dynamic (SD) model. The description of the SD model in line 196 (“a computer-aided approach for stock and flow diagrams”) seems to be not appropriate.

6.  Please revise the equations provided in the MS.

-   In Eq. 3 the unit of 3600 is missing.

- In Eqs. 4 and 6 the units of the terms on the left-hand side and on the right-hand side are different.

- In Eq. 7. m_air "specific mass of air" should be called as air density.

7.  The Conclusions are not fully supported by the results.

In lines 406-408 the Authors stated the followings: "Other two scenarios that also include 50% of the window opening (with the green tempered glass and EPS wall inclusion) have lower percentages, but with no significant difference, even comparing the temperature peak value." In my opinion, this means that there is no significant difference whether green tempered glass is used or not. However, according to lines 426-429 it seems to me that the usage of green tempered glass causes significant differences in the results since the Authors concluded that: "This research demonstrates that natural ventilation and the use of constructive elements of low complexity and wide availability in the market (tilting window with up to 100% opening, green tempered glass, and EPS wall) contribute to the thermal comfort of a residential room."  This seems like a contradiction to me. Please explain it.

8. In the MS, the configuration 50% Window opening + EPS is not used as a scenario. Please explain why this scenario was not examined? It may support the conclusions of the MS if this scenario may perform better than the scenario 50% Window opening but it may perform worser than the scenario 50% Window opening + EPS + Glass.

9.  Since only one day was analyzed in the MS, it raises the question how robust the results are? The Authors stated that the selected day presents “a severe but representative condition for the thermal comfort simulation”. However, is the time course of the solar radiation – with the three local maxima shown in the left-hand side of Fig. 5 – also representative? To what extent do the results depend on the solar radiation?

10. The source of the solar radiation data is missing. Please add a reference to it.

11. In line 37 the following is written: "wood and other renewable resources such as hydro, solar, wind, and biomass". Please explain why wood is not considered as a type of biomass.

12. It is not obvious whether Figs. 1 and 2 are the own works of the Authors or not. Please clarify it and add that information to the subtitles of the figures.

13. In lines 348-352 please specify what "graph curve" and "box" are mentioned since there are no graphs and boxes in Fig. 4. Furthermore, what is the meaning of "Known values such as weather, [...]"? Are the Authors referring to the meteorological variables e.g., temperature and wind?

Minor comments:

1. The study area (Baltimore Residential in Uberlândia city) cannot be located. I suggest the Authors to provide a map.

2.  Sources [3] and [33] are the same.

3.  In line 117 the abbreviation of System Dynamics (SD) is introduced, so the abbreviation SD can be used in line 183.

4. The list in lines 191-193 seems to be connected to the 1st sentence of the section Materials and Methods. Consequently, the list should be placed right after the 1st sentence.

5. Please name the softwares which were used to create the figures of the MS.

6. In line 208 I don't think that the expression "diverse parameters" is appropriate.

7.  In line 228 there is "façade".

8. In line 265 there is "corrected wind speed" instead of average corrected wind speed.

9. In lines 312 and 332 there is punctuation mark (dot) instead of product sign.

10.  Please define "mixed material scenarios simulation" in line 368.

11. In the section References the digital object identifiers (DOIs) are completely missing. Please insert them in cases of all references where the DOIs are available.

Reviewer 2 Report

This paper makes use of existing models and methods to evaluate a single room using simulation. The experimental settings are presented in detail, but the simulation is too simple and not technically sound. Therefore, I encourage the authors to improve their simulation using more room structures and window settings.

1. It is not clearly shown that how the room structure will affect the airflow and thermal comfort. Opening windows for more airflow is a common knowledge.

2. ASHRAE have proposed the PMV model for evaluating thermal comfort, why not using the PMV index?

3. How the location and orientation of the windows will affect the indoor temperature?  Inappropriate orientation of windows may lead to unreasonably high indoor temperature.

4. The author doesn't give any implications for problems like 1) how to determine the size and the locations of the window according to the room structure; 2) how to determine the distance between two buildings so that increased airflow and reduced sun heat can be achieved.  Such information is important to the designers.

Round 2

Reviewer 1 Report

The Manuscript has certainly improved. New references were added to it and the questions of the reviewer were answered.

Some minor comments:

ASHRAE 55 is mentioned first on page 11, so its meaning (American Society of Heating, Refrigerating and Air-Conditioning Engineers) should be relocated from page 14 to here.

There are double commas in the line above Eq. 3.

In the paragraph between Eqs. 3 and 4 there is "Error! Reference source not found.".

There is a missing line break at the beginning of Section 4.

Reviewer 2 Report

The presentation of the paper is improved, but the major concern is not addressed. There is no comparison between different architectural patterns of the houses, and the influence of the location and size of the window are not examined in the simulation. The authors should at least improve their discussion with these issues. Otherwise, the study provides limited information for the readers. Nevertheless, I think this paper may deliver some implications for policy and practice, which may contribute to sustainability.

Round 3

Reviewer 2 Report

Thank you for the effort in revising the manuscript, and I have no more comment.